# Artificial Intelligence to Early Predict Liver Metastases in Patients with Colorectal Cancer: Current Status and Future Prospectives

**DOI:** 10.3390/life13102027

**Published:** 2023-10-09

**Authors:** Pasquale Avella, Micaela Cappuccio, Teresa Cappuccio, Marco Rotondo, Daniela Fumarulo, Germano Guerra, Guido Sciaudone, Antonella Santone, Francesco Cammilleri, Paolo Bianco, Maria Chiara Brunese

**Affiliations:** 1HPB Surgery Unit, Pineta Grande Hospital, Castel Volturno, 81030 Caserta, Italy; biancopaolo@virgilio.it; 2Department of Clinical Medicine and Surgery, University of Naples “Federico II”, 80131 Naples, Italy; 3Department of Medicine and Health Sciences “V. Tiberio”, University of Molise, 86100 Campobasso, Italy; giggiterry1393@gmail.com (T.C.); m.rotondo1@studenti.unimol.it (M.R.); d.fumarulo@studenti.unimol.it (D.F.); germano.guerra@unimol.it (G.G.); guido.sciaudone@unimol.it (G.S.); antonella.santone@unimol.it (A.S.); mariachiarabrunese@libero.it (M.C.B.); 4Gastroenterology Unit, A. Cardarelli Hospital, 86100 Campobasso, Italy; francesco.cammilleri@asrem.org

**Keywords:** artificial intelligence, liver neoplasm, liver metastases, early prediction, colorectal liver metastases

## Abstract

Background: Artificial Intelligence (AI)-based analysis represents an evolving medical field. In the last few decades, several studies have reported the diagnostic efficiency of AI applied to Computed Tomography (CT) and Magnetic Resonance Imaging (MRI) to early detect liver metastases (LM), mainly from colorectal cancer. Despite the increase in information and the development of different procedures in several radiological fields, an accurate method of predicting LM has not yet been found. This review aims to compare the diagnostic efficiency of different AI methods in the literature according to accuracy, sensibility, precision, and recall to identify early LM. Methods: A narrative review of the literature was conducted on PubMed. A total of 336 studies were screened. Results: We selected 17 studies from 2012 to 2022. In total, 14,475 patients were included, and more than 95% were affected by colorectal cancer. The most frequently used imaging tool to early detect LM was found to be CT (58%), while MRI was used in three cases. Four different AI analyses were used: deep learning, radiomics, machine learning, and fuzzy systems in seven (41.18%), five (29.41%), four (23.53%), and one (5.88%) cases, respectively. Four studies achieved an accuracy of more than 90% after MRI and CT scan acquisition, while just two reported a recall rate ≥90% (one method using MRI and CT and one CT). Conclusions: Routinely acquired radiological images could be used for AI-based analysis to early detect LM. Simultaneous use of radiomics and machine learning analysis applied to MRI or CT images should be an effective method considering the better results achieved in the clinical scenario.

## 1. Introduction

In the last few decades, multidisciplinary teams of surgeons, anesthesiologists, oncologists, and radiologists have collaborated to increase the overall survival (OS) and disease-free survival (DFS) of oncologic patients [1,2,3,4].

Although the radical change in surgical context with the advent of innovative techniques and surgical approaches improved short- and long-term outcomes in various fields of abdominal cancer [5,6,7,8,9,10,11,12,13], no predictive marking of metastases has been identified.

In recent years, artificial intelligence (AI) has become an integral part of healthcare through the association of algorithms, machine learning, computers, and data sciences [14,15,16,17]. Furthermore, AI-based analysis is increasing thanks to the ability to quantify imaging aspects unobservable to the human eye for the early prediction of tumor or metastatic spread both in Computed Tomography (CT) and in Magnetic Resonance Imaging (MRI) images [18,19,20,21,22,23,24].

In this clinical scenario, AI is also applied to patients affected by colorectal cancer (CRC) [21,25]. CRC is one of the main cancers worldwide, and about 40–50% of patients will develop colorectal liver metastases (CRLM) either during oncological follow-up, after primary tumor resection, or during postoperative chemotherapy regimen administration [1,9,26,27]. Therefore, more than 20% of CRC patients have distant metastases at the time of diagnosis [28], registering a 5-year OS of about 15% compared to 80–90% in patients with local disease. The liver represents the main metastatic site because of its portal-systemic circulation. Several surgical approaches are proposed to treat patients with synchronous resectable colorectal cancer and liver metastases: the staged approach, which includes primary tumor resection followed by chemotherapy and then LM resection; a combined approach with primary and metastatic disease resection; and the “liver first” approach [1].

These options are suitable for about 25% of patients. In this clinical scenario, an extensive analysis of liver texture—through matrices, filters, and transforms—could allow for early prediction of CRLM-identifying metastasis features during imaging studies [25,29,30]. So, early CRLM treatment could be performed [9,31]. Furthermore, the widespread use of AI applications is also reported in predictive models of OS after surgery and/or chemotherapy and as a prediction of post-operative outcomes [30,32,33].

However, while different AI methods are described in the literature [14], the most effective and safe techniques to predict CRLM are still debated [21].

In the precision-medicine era and personalized oncological approaches epoch, it is mandatory to obtain early detection of metastatic disease in order to improve OS, DFS, and quality of life [17,23,34,35].

This review aims to compare the diagnostic efficiency of different AI methods reported in the literature, according to accuracy, sensibility, precision, and recall in identifying early LM.

Additionally, our secondary endpoints include a clarification of AI methods in the treatment of patients affected by CRLM through an extensive analysis of the advantages and disadvantages of AI in surgery.

## 2. Material and Methods

Systematic research was carried out on PubMed, Cochrane Library, Kaggle and Data.gov on 2 June 2022, and the studies considered have already been published in English.

Keywords included in the search were: (artificial intelligence) OR (machine learning) OR (radiomics) OR (machine learning) AND (liver OR hepatic) AND (metastasis) OR (metastases). Our review was conducted according to the Preferred Reporting Items for Statistics Reviews and Meta-Analysis (PRISMA) statement and Strengthening the Reporting of Observational Studies in Epidemiology (STROBE), which were used to conduct the review [36,37]. Duplicates were screened and removed.

The main feature included in our review concerns the use of AI algorithms, followed by machine learning, deep learning, and radiomics algorithms to predict the development of CRLM, analyzing the primary tumor (colon or rectum) differences.

Additionally, studies that included ≤20 patients, randomized clinical trials, and papers not written in English (e.g., Chinese) were excluded. Furthermore, we excluded literature focusing on predicted survival rate analysis by AI or other topics beyond the scope of this study, as well as preclinic or non-human studies, case reports, editorials, and reviews. Additionally, studies that analyzed CRLM and other tumors (hepatocellular carcinoma, intrahepatic cholangiocarcinoma) simultaneously were also excluded. We selected the largest cohort of patients, covering a wide range of cases, from multiple papers with superimposable data published by the same research groups.

All articles were selected by two authors (MC, TC) and subsequently managed by a third author (PA).

Data extracted are: algorithms used for CRLM prediction, number of patients included in the study, primary tumor location, CT or MRI images, machine training type (e.g., training set, test set, and validation set), accuracy, specificity, recall, and precision rates.

## 3. Results

A total of 336 articles were identified in PubMed and, after skimming them, only 17 were found to be suitable for our analysis. No studies were available in the Cochrane Library, Kaggle, and Data.gov databases. Figure 1 summarizes the number of papers and related exclusion criteria after title, abstract, and full-text analysis.

The total number of patients was 14,475. In total, 3612 (24.95%) patients were affected by rectal cancer and 786 (5.43%) by colon cancer. In 11 cases, the stratifications were not performed; 9816 (67.81%) patients with CRC were included. Furthermore, in 206 (1.42%) patients, the primary tumor site was not specified, and in 55 (0.39%) cases, the analysis was conducted on HCC or focal nodule hyperplasia. Only 517 (3.94%) patients underwent chemotherapy [38,39]. In total, 10 (58.82%) out of 17 studies carried out AI-based analysis using diagnostic images obtained by CT scan, 3 (17.65%) by MRI, and in 1 (5.88%) cohort, it was not specified. In 3 (17.65%) cases, both CT and MRI were used [29,39,40].

These findings are summarized in Table 1. A report of the baseline characteristics of patients and AI methods is presented in Table 1. Four studies reported the time of metastasis detection from primary tumor diagnosis [18,21,40,41]: from 3 to 48 months, patients developed metastases. Images of 14,269 features were extracted. In detail, 7947 patients’ images were used to train the machine and were therefore useful for the training set, while 645 patient images were used for the test set, and images of 2082 patients were used for the validation set. Furthermore, 80 patients were included in cross-validation set.

Additionally, the accuracy, precision, specificity and recall rate are listed in Table 2.

Figure 2 represents Regions of Interest (ROI) of apparently healthy liver parenchyma in patients affected by primary colorectal cancer without radiological evidence of liver metastasis.

## 4. Discussion

This review screened 17 research papers evaluating AI-based analysis to predict CRLM. Most papers have been published in the last three years, underscoring the progressive spread of radiomics in the medical sciences [50,51]. AI includes all available applications of technologies in various medical fields [23,33,52,53]: adequate training of AI could potentially perform early diagnosis of unknown diseases, reducing clinicians’ biases and time to treatment [21,54,55,56].

### 4.1. Colorectal Cancer and Liver Metastases

Between 40% and 50% of patients affected by CRC will develop CRLM during follow-up [57,58]. Nevertheless, only 25% of these patients are eligible for surgical resections. Due to the variability in surveillance strategies for patients affected by CRC, there is no worldwide consensus on patients’ management after primary tumor resection; however, surgery remains the preferred treatment for CRLM [28,41,59,60,61,62,63,64,65,66].

In the last few decades, it has been widely demonstrated that minimally invasive liver surgery achieves intraoperative and postoperative results comparable to those obtained by open surgery both in terms of complications and overall survival [27,67,68,69,70,71,72,73,74,75,76], especially when R0 resection is performed. This has also been reported by randomized clinical trials [77,78].

For this reason, the early detection of metastases is crucial to improving survival outcomes [21,28,41].

The most frequently used imaging method to detect CRLM is a CT scan with intravenous contrast [79,80]. In the literature, CT showed sensitivity and specificity rates of up to 85% and 97%, respectively [81].

Considering that CRLM development occurs within 12 months after surgical CRC, the occurrence of metastases may be due to inadequate image analysis of liver texture or less intense screening protocols and adjuvant chemotherapies [21,41].

In the literature, many authors have described micro-environmental changes in healthy livers in patients affected by CRC [18,79,82]. These findings are also present in the early phase of liver metastases, but they are not visible to the human eye [18]. The prediction could be performed thanks to different textures of hepatic parenchyma: in CT and MRI images, the metastases are shown by a higher grey level entropy if compared to healthy liver parenchyma [83]. This could be explained by dissimilar cell clones, necrosis, and, mainly, by the chaotic tumor vasculature [83]. However, the variability and heterogeneity of metastases contribute to the difficult early diagnosis of CRLM [21].

The portal-venous phase images are more predictable in the diagnosis of CRLM due to the hypodense texture and whether or not the peripheral rim is improved. However, in a non-neglectable percentage of cases, patients affected by CRC present liver nodules with radiological characteristics superimposable to CRLM but not defined as metastases due to their small size. In these cases, an MRI, biopsy, or repeat CT could be performed to define the nodule nature. It should be emphasized that more than 15% of hypodense nodules will evolve into malignant tumors [32]. The effort to detect early CRLM, due to operator variability and the costs of CT or MRI, is rewarded by better surgical outcomes obtained in patients with minor liver resections compared to major resections [84].

### 4.2. Artificial Intelligence

In the last few years, CT and MRI texture analysis of the liver have been evaluated to early predict metachronous CRLM [22,30,32,41], considering the findings obtained by AI in other medical fields [15,19,56,85,86]. For these reasons, different AI-based analyses have been proposed to predict CRLM, including machine-learning (ML) and deep learning (DL) methods. Indeed, ML represents a subset of AI that, through statistical algorithms, makes it possible to acquire skills through experience [14,87,88]. Additionally, DL is a multilayered, complex ML platform that can train itself using neural networks to make accurate predictions [14]. The differences among AI, ML, and DL are summarized in Figure 3.

In 2020, Lee et al. [45] published a retrospective study analyzing liver imaging achieved by abdominal CT images of patients with stage I–III CRC using convolutional neural networks (CNN). CNN was used to obtain imaging features from the liver parenchyma. Through a dimension reduction, the authors designed multiple prediction models for 5-year metachronous liver metastasis, also using combinations of clinical variables such as age, sex, tumor, and nodal stages and top principal components (PCs) with logistic regression classification.

Simultaneously, Li et al. [47] compared 50 patients affected with CRLM and 50 patients without CRLM. The authors developed a quantitative analysis platform for medical images, defined as the Radiomics Intelligent Analysis Toolkit (RIAT), to assist radiologists. Thanks to the RIAT, which matched radiological images and clinical information, they were able to predict CRLMs.

In 2021, Taghavi et al. [18] used clinical features and radiomic findings such as liver contrasted-enhanced venous phase CT, primary tumor site, tumor stage, nodal stage, CEA levels at diagnosis of CRC, age, sex, and chemotherapy regimen (neoadjuvant and/or adjuvant) of 91 patients to obtain an accurate method to predict CRLM development in patients during follow-up. Indeed, the authors detected 24 metachronous CRLM.

Despite several studies published in the literature, the quality of the evidence was low in most cases. Our review included 14,475 patients affected by CRC who underwent AI analysis to predict CRLM from 17 studies, using different engineering techniques and radiological images. The analysis was conducted to outline favorable scenario and the most effective AI method. The accuracy, defined as the rate of correctly classified cases (true positive + true negative divided by true positive + true negative + false positive + false negative), has been used to evaluate the study results.

In our review, the Formal method used by Rocca et al. [21], the neural network and fuzzy genetic algorithm used by AmirHosseini et al. [43], and the radiomics and machine learning analysis used by Granata et al. [29] represent the most accurate methods. The Formal method is based on Milner’s algebra and is used to obtain a mathematical model of patients’ characteristics [21]. Consequently, a multidisciplinary team defines the quantitative characteristics of a state of health to find common features in the same class of patients and automatically recognize healthy and unhealthy liver images. Thanks to formal methods applied to CT images, Rocca et al. [21] achieved an accuracy rate >90% in a limited series of patients (30 patients). Nevertheless, the study presents many limitations: (1) The regions of interest (ROIs) evaluated were manually defined and therefore operator-dependent; (2) patients were selected from a single-center with a single CT type. In conclusion, The Formal method should be used to analyze more numerous cohorts of patients.

On the other hand, AmirHosseini et al. [43] used the neural network to create a computer model inspired by the biological nervous system, and fuzzy genetic algorithm, an ordering sequence of instructions designed with fuzzy logic-based tools, and obtained an accuracy rate of 99% analyzing CT images of 125 patients. The authors validated these findings by comparing AI results with radiologists’ diagnoses.

A subclass of machine learning, deep learning, was used by Goehler et al. [46] to predict CRLM in 64 patients. Deep learning analyzes raw data and performs a detection of CRLM without human intervention using algorithm sets (generally artificial neural networks). Through this method, the authors obtained 88% accuracy after consecutive MRI scans. The authors obtained a sensitivity of 0.85 and a specificity of 0.92.

Moreover, a huge number of patients who underwent CT and MRI scans were analyzed by Granata et al. [29]. Indeed, the authors studied patients in T2W space, arterial and portal venous phases, and enhanced MRI often achieved an accuracy rate >80%. It should be underlined that the portal venous phase accuracy value was 91%. Nevertheless, the study presents many limitations, including the chemotherapy impact and the absence of a validation external dataset.

One drawback in terms of analysis is the accuracy rates, which are not available for the extensive studies published by Lee, Liang, and Kim et al. included in our review [40,45,49].

However, as reported in our previous paper, another key value is represented by recall rate, which indicates the assignment completeness to the subset of the positives by evaluating the patients’ number, appropriately classified as positive on the total of positives [21].

The studies evaluated in the literature report heterogeneous values, also considering internal and external validation tests used during the AI application to CT and MRI images [89,90]. This finding does not allow us to compare and argue about the results obtained by different methods. Nevertheless, Granata et al. [29] obtained the greatest rate when compared to other studies. In addition to those previously reported, these findings suggest that the portal venous phase represents a significant phase for identifying CRLM patterns and predicting their development. Although, due to the limited availability of data, it is not possible to evaluate the effectiveness of manual or automatic segmentation of scan images, as reported in the literature [21], which only represents user-dependent data.

### 4.3. Chemotherapy Regimen

In addition to the surgical approaches that have radically changed colorectal surgery and perioperative outcomes in patients affected by the early stages of CRC [91,92,93,94,95], the combination of cytoreductive surgery, chemotherapy, and hyperthermic intraperitoneal chemotherapy (HIPEC) in cases of peritoneal metastases achieved a 5-year survival rate of up to 30% [96].

After an extensive literature review, we included chemotherapy data just for two studies (Table 2). To our knowledge, chemotherapy could impact the size of metastases, especially in cases of micro-environment liver texture. Furthermore, regarding chemotherapy, some authors reported the safety application of AI methods to early evaluate the chemotherapy response of patients [79]. Nevertheless, it could be established that radiomics and AI-based analysis may be preferred compared to the standard biomarkers used in patients affected by CRLM.

As described by Devoto et al. [41], a poorer survival rate was associated with a lower heterogeneity in liver features. It is a key point for prospective studies that could investigate the prognosis based on contrast-enhanced CT or MRI. However, in the last few decades, a worldwide consensus on simultaneous resection of CRC and synchronous CRLM has been reported in the literature [9,97,98,99,100]. The benefits of simultaneous resections could be strengthened through AI-based analysis of liver texture: an early detection of non-healthy microenvironments in patients affected by CRC, followed by liver and primary tumor resection, is linked to suitable oncological outcomes [101].

In a recent meta-analysis, Komici et al. [102] evaluated the prevalence and impact of frailty in HPB surgery: Frailty was found in about 40% of patients affected by liver cancer, including CRLM. Indeed, comorbidities and frailty should be managed in elderly patients to predict inter-individual variability [103,104]. It is a crucial point: an early detection of CRLM could improve long-term outcomes in this subset of patients, considering the mortality rate reported for major hepatectomies [71,105].. Future perspectives are linked to AI applications to predict liver failure or future liver remnants in the preoperative evaluation of patients affected by huge liver tumors [106,107].

### 4.4. Current Status and Future Perspectives

AI should assist radiologists during CRLM identification through the analysis of medical images. Therefore, AI algorithms can automatically segment the liver, achieving a preoperative study of future liver volume and the parenchyma resection for the surgeons’ team during the planning of a treatment strategy. Furthermore, AI-based simulations and educational tools can help train young healthcare professionals, in both surgical and oncological fields, to diagnose and treat CRLM effectively.

Several studies have reported that AI tools can also predict outcomes and recurrence risk based on various clinical and genetic factors [24,38,53,108]. In this clinical scenario, AI should guide the physician during short- and long-term follow-up and surveillance of patients affected by CRC and LM.

Patients declared unfit for surgery could benefit from AI-driven therapies. AI could identify potential drug therapies by analyzing genetic and molecular data [109].

This future finding will contribute to worldwide precision medicine: recent knowledge about the spatially varying perfusion coefficient obtained through dynamic contrast-enhanced MRI or Doppler ultrasound could be integrated into image analysis to evaluate perfusion heterogeneity in biological tissues [109,110]. In fact, tumor vascularization arises through several biological processes. These processes occur due to a huge variability of factors and pathways and involve progenitors or cancer stem cells. Early identification of these processes could unmask the growth of metastatic lesions within the hepatic parenchyma [110,111,112].

### 4.5. Limitations

Our aim was to give an overview of AI-based analysis in the early prediction of CRLM. However, the present review includes some limitations. The papers analyzed present wide-ranging variability in sample size, comparing small series of cases to huge cohorts. Furthermore, an extensive heterogeneity of AI methods, including dissimilar software applications, does not contribute to identifying the best AI-based analysis for early prediction of CRLM. Rigorous testing and validation of AI algorithms in a clinical setting are essential but time-consuming and expensive.

Therefore, the different use of CT and MRI images does not allow a comparison between liver texture analysis in healthy and unhealthy patients.

Furthermore, a key point concerning ethical considerations regarding computed clinical investigations should be made. Healthcare data are sensitive and valuable, making them a prime target for cyberattacks, requiring robust security measures to protect patient information and maintain data integrity.

These aspects should be evaluated, as well as technology infrastructure and staff training, during AI cost analysis. While AI holds great promise for revolutionizing healthcare, these limitations must be carefully considered and addressed to maximize its benefits and ensure safe and equitable use in medical applications. Ongoing research and collaboration between AI experts, healthcare professionals, and regulatory bodies are essential to overcome these challenges.

In the coming years, AI systems may be used to make medical decisions or recommendations, determining a lack of accountability in cases of errors or adverse outcomes. Questions about who is ultimately responsible for AI-generated decisions are still being explored.

Considering the fast-evolving field, these limitations should be overcome by future case series and randomized clinical trials.

## 5. Conclusions

This review demonstrates the presence of CRLM in healthy liver texture analysis. The complexity of metastasis images could be analyzed through AI methods to investigate the heterogeneity and variability of a subset of metastatic cells not visible to the naked eye. These fields of medical and surgical treatment of patients affected by CRC should be an efficient and safe diagnostic process during radiological investigations, resulting in potential benefits for patients and healthcare systems in terms of timing and costs. Moreover, AI should be a potential tool for prognostic and outcome prediction, especially during patient follow-up. In this context, AI could perform a stratification of prognosis in terms of OS and DFS to drive the best management of patients affected by CRLM. These promising findings should be supported by further prospective studies on AI methods applied to HPB surgery.

## Figures and Tables

**Figure 1 life-13-02027-f001:**
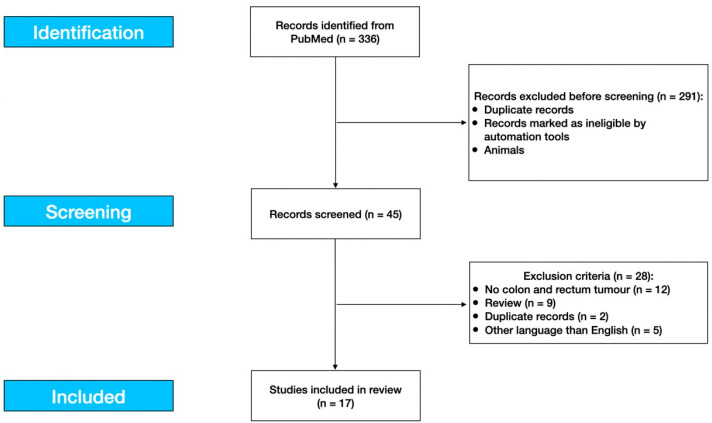
Study flow diagram.

**Figure 2 life-13-02027-f002:**
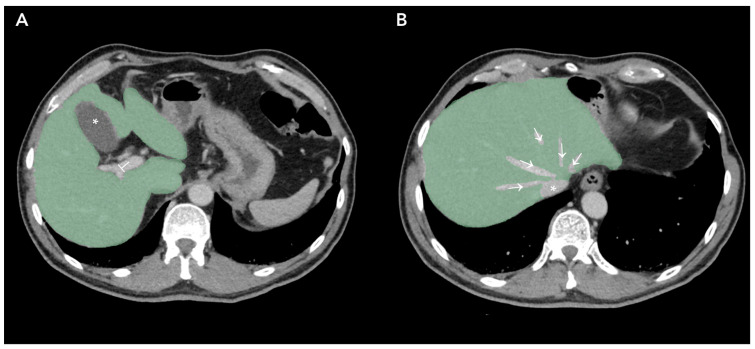
Regions of Interest obtained by Contrasted-Enhanced Computed Tomography slices of patients affected by Colorectal Cancer. (**A**), liver parenchyma marked green, gallbladder (*) and portal vein (⊥) exclusion; (**B**), liver parenchyma marked green, suprahepatic veins (→) and inferior vena cava (*) excluded.

**Figure 3 life-13-02027-f003:**
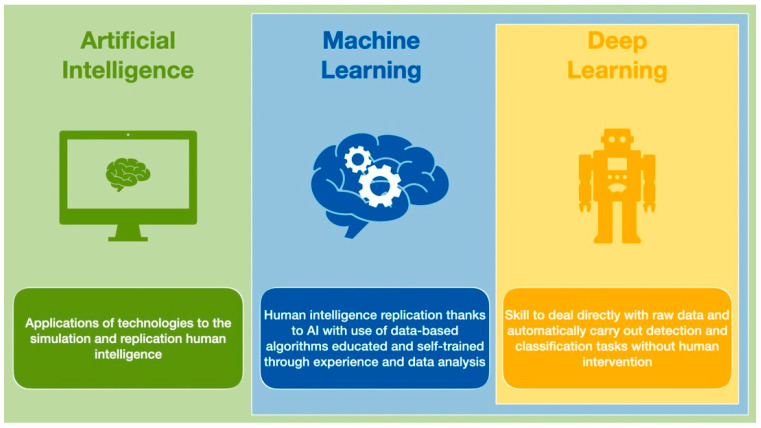
Artificial intelligence, machine learning and deep learning definitions. AI: Artificial Intelligence.

**Table 1 life-13-02027-t001:** Articles included in our review: AI methods, primary tumor, location and imaging types.

Author	Year	Method	N. of Patients	Primary Tumor	Patients Division for Model Checker & Validation Set	Chemotherapy (Yes/No)	Imaging	Time of Metastases Detection from Primary Tumor Diagnosis (Months)
Biglarian et al. [42]	2012	Artificial neural networks (ANNs) & Logistic regression (LR)	1007 (786 CC, 204 RC)	CRC	705 patients for training set and 302 patients for validation set	NA	NA	NA
Shu et al. [39]	2019	Radiomics nomogram	194	RC	135 independent cohorts for training set and 59 patients for validation set	No	MRI + CT	NA
Liang et al. [40]	2019	Machine learning: Support vector machine (SVM) & LR	3087	RC	NA	NA	MRI + CT	<12 months
AmirHosseini et al. [43]	2019	Neural network and Fuzzy genetic algorithm	125	NA	NA	NA	CT	NA
Liu et al. [44]	2020	Radiomics nomogram	127	RC	88 patients for training set and 39 for validation set	NA	CT	NA
Han et al. [34]	2020	Machine learning: Classification model (Decision tree)	107	CRC	NA	NA	MRI	NA
Lee et al. [45]	2020	Convolution neural network (CNN)	2019	CRC	1413 patients for training set and 606 for test set	NA	CT	NA
Goehler et al. [46]	2020	Deep learning model	64	NA	45 patients for training set, 19 for test set, 20% of training data was split to validation set	NA	MRI	NA
Taghavi et al. [18]	2020	Machine learning based radiomics model	91	CRC	70 patients for training set, 21 for validation set	NA	CT	24 patients developed metastases before 24 months after primary staging
Li et al. [47]	2020	Radiomics intelligent analysis toolkit (RIAT) and LR	100	CRC	80 patients for cross validation set and 20 for test set	NA	CT	NA
Lee et al. [48]	2021	Deep learning model	502	CRC	NA	NA	CT	NA
Kim et al. [49]	2021	Deep learning model	6526	CRC	5129 patients for training set and 1397 for validation set	NA	CT	NA
Stollmayer et al. [22]	2021	Densely connected convolutional neural networks (DenseNets)	69	14 CRC (42 Focal nodule hyperplasia and 13 HCC)	NA	NA	MRI	NA
Rocca et al. [21]	2021	Formal methods	30	CRC	NA	NA	CT	3–48 months
Li et al. [38]	2022	Machine learning: Classification model	323	CRC	171 patients for training set, 77 for internal validation set and 75 for external validation set	No	CT	NA
Granata et al. [29]	2022	Radiomics & Machine learning analysis	81	CRC	51 patients for training set, 30 for external validation set	NA	MRI + CT	NA
Devoto et al. [41]	2022	Textural analysis	23	CRC	NA	NA	CT	<7 months

CRC, colorectal cancer; CC, colon cancer; RC, rectal cancer; NA, not available; MRI, Magnetic Resonance Imaging; CT, Computed Tomography; ANN, Artificial neural networks; SVM, Support Vector Machine; LR, Logistic regression.

**Table 2 life-13-02027-t002:** Articles included in our review: dataset type, number of features and results obtained according to artificial intelligence used.

Author	Dataset	N. of Features	Accuracy	Specificity	Precision	Recall	AUROC	AUAFROC	CI
Biglarian et al. [42]	Training set Validation set	NA	NA	RC with ANN: 85.7%, CC with ANN: 91.4%, CC with LR: 92.3%	NA	RC with ANN: 44.4%, CC with ANN: 48.6%, CC with LR: 32.4%	NA	NA	ANN: 0.812 LR: 0.779
Shu et al. [39]	Training set Validation set	328	Training set: 92.1%, Validation set: 91.2%	NA	NA	NA	Training set: 85.7%, Validation set: 83.4%	NA	Accuracy training set: 0.862–0.961, Accuracy validation set: 0.809–0.970
Liang et al. [40]	5-fold cross validation *	35	LR: 80% SVM: 72%	LR: 76% SVM: 69%	NA	LR: 83% SVM: 76%	LR: 87% SVM: 83%	NA	LR: 0.730–0.880, SVM 0.650–0.840
AmirHosseini et al. [43]	Training set Test set	NA	99.24%	Neural network: 85.7%, Fuzzy GA: 100%	NA	Neural network: 81.8%, Fuzzy GA: 96.67%	98.32%	NA	Neural network: 0.744–0.925, Fuzzy GA: 0.885–1.000
Liu et al. [44]	Training set Validation set	866	Validation set: 89.66%	Validation set: 93.65%	NA	Validation set: 79.17%	Validation set: 86.6%	NA	Validation set: 0.770–0.963
Han et al. [34]	Training set, Internal validation set, External validation set	182	Internal validation set: 95.2%, External validation set: 78.8%	Internal validation set: 70%, External validation set: 34.5%	NA	Internal validation set: 95.2%, External validation set: 100%	Training set: 97.1%, Internal validation set: 90.9%, External validation set: 90.5%	NA	NA
Lee et al. [45]	5-fold cross validation *	4096	NA	NA	NA	NA	74.7%	NA	NA
Goehler et al. [46]	X-fold cross validation *	NA	88%	92%	92%	85%	NA	NA	Recall: 0.770–0.930, Specificity: 0.870–0.960
Taghavi et al. [18]	Training set Validation set	101	NA	NA	NA	NA	Training set: 93%, Validation set: 86%	NA	Training set: 0.910–0.950, Validation set: 0.850–0.870
Li et al. [47]	5-fold cross validation *	210	NA	Test set: 91%, Validation set: 75%, Cross validation set: 84%	NA	Test set: 78%, Validation set: 85%, Cross validation set: 81%	Test set: 89.9%, Validation set: 86%, Cross validation set: 90.6%	NA	Test set: 0.761–1.000, Cross validation set: 0.840–0.971
Lee et al. [48]	Training set Validation set	NA	71.66%	72.78%	NA	70.47%	84.10%	NA	NA
Kim et al. [49]	Training set and Validation set	NA	NA	22.22%	NA	87.50%	NA	0.631 (0.520–0.737)	NA
Stollmayer et al. [22]	Training set, Test set and Validation set	NA	NA	2D model: 100%, 3D: 95%	NA	2D model: 80%, 3D: 70%	Test set 2D model: 96%, 3D: 90.5%	NA	Test set 2D model: 0.879–1.000, 3D: 0.789–1.000
Rocca et al. [21]	NA	22	93.3%	100%	100%	77.8%	NA	NA	NA
Li et al. [38]	Training set Validation set	1288	Validation dataset 1: 58.4%, Validation set 2: 65.3%	Validation dataset 1: 48.2%, Validation set 2: 65.4%	NA	Validation dataset 1: 85.7%, Validation set 2: 65.2%	Validation dataset 1: 79%, Validation set 2: 72%	NA	Validation dataset 1: 0.680–0.870 Validation set 2: 0.600–0.820
Granata et al. [29]	Training set Validation set	851	Validation set T2W SPACE: 86%, Arterial Phase: 89%, Portal Phase: 91%, EOB Phase: 80%	Validation set T2W SPACE: 86%, Arterial Phase: 91%, Portal Phase: 96%, EOB Phase: 100%	NA	Validation set T2W SPACE: 86%, Arterial Phase: 85%, Portal Phase: 81%, EOB Phase: 67%	Validation set T2W SPACE: 88%, Arterial Phase: 96%, Portal Phase: 99%, EOB Phase: 95%	NA	NA
Devoto et al. [41]	NA	NA	NA	63.6%	NA	83.3%	75%	NA	NA

AUROC, ROC curve; AUAFROC, Area Under the Alternative Free-Response Receiver Operating characteristic Curve; CI, Concordance Index; CRC, colorectal cancer; CC, colon cancer; RC, rectal cancer; NA, not available; ANN, Artificial neural networks; SVM, Support Vector Machine; LR, Logistic regression; MRI, Magnetic Resonance Imaging; CT, Computed Tomography; * X-fold cross validation: Training set, test set and validation set.

## Data Availability

The datasets used and/or analyzed during the current study are available from the corresponding author on reasonable request.

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
