# Peer review of "Artificial Intelligence to Early Predict Liver Metastases in Patients with Colorectal Cancer: Current Status and Future Prospectives"

_life, 2023, doi:10.3390/life13102027_

Round 1

Reviewer 1 Report

The manuscript is a good review of what have been pubblished on regard the prediction of liver metastases.

I suggest to the authors to report another table where it could be reported a comparison between the starting number of dataset considered in each selected study, the number of texture features extracted from the dataset and , if possibile, the charcteristics of the dataset analyzed (i.e. those related to the CT or MRI images considered). This could help the reader to have a vision of what was performed and possible ideas of investigation. Moreover, based on this table, the authors could say something about possible pros and cons of the previously perfomed analysis ( oversampling or undersampling occuring in the dataset used by previous studies).

Finally, take a look on typo errors in the manuscript (such as at line 58, 249)

Author Response

  1. The manuscript is a good review of what have been published on regard the prediction of liver metastases. I suggest to the authors to report another table where it could be reported a comparison between the starting number of dataset considered in each selected study, the number of texture features extracted from the dataset and , if possibile, the characteristics of the dataset analyzed (i.e. those related to the CT or MRI images considered). This could help the reader to have a vision of what was performed and possible ideas of investigation.

Thank you for your suggestion. We emphasized the dataset considered in studies included in our review (Table 1).

  1. Moreover, based on this table, the authors could say something about possible pros and cons of the previously perfomed analysis (oversampling or undersampling occuring in the dataset used by previous studies).

We added “Current status and future prospectives” section in discussion.

  1. Finally, take a look at typo errors in the manuscript (such as at line 58, 249).

We reviewed all paper’s sections in order to improve the study quality. Thank you for your precious time.

Reviewer 2 Report

Peer Review Report

Manuscript ID: Life-2626085

Title: Artificial Intelligence to early predict liver metastases: a narrative review of 14,475 patients

The study “Artificial Intelligence to early predict liver metastases: a narrative review of 14,475 patients” by authors Avella et al. lies within the Journal scope of Life. There are several suggestions for authors to incorporate before recommending or reconsider the work for publication. The authors fail to formulate a Research Question? The authors focus was little shifted in writing. The authors should pay attention to all aspects and given comments in next submission and ensure that all points should be adequately addressed.  

1. The title of the study should be revised. There should not be any mention of quantitative number in the title. Furthermore, “anarrative” should be “a narrative” for present title.

2. The Abstract section should be rewritten. There are several long unstructured sentences that needs revision. For example:

Due to higher information contents and 19 different procedures, it was not founded the greatest method to predict LM so far. Revise Abstract section.

3. Proof-read of entire manuscript is required. For example in Introduction section there is no spacing between words or extra spacing within two words: (Lines 37-39), Line 42, Line 52, Line 62, Line 96, Line 130, Line 173

In the last decades, multidisciplinary teams of surgeons, anesthesiologists, oncologists, and radiologists have beencollaborated to increase the oncologic patients’ Overall Survival (OS) and Disease-Free Survival (DFS)

What do you mean by trough matrices?

Line 62: theadvantages

Line 173: Plural: For these reason

4. The Introduction section should be rewritten. In current submission, the provided Introduction is very weak.

5. Section 2 name should only be “Materials and Methods” and not Materials and Methods Literature Research.

6. Criterion used to hunt for studies should be better described. Also, the authors are advised to pay attention on punctuation. For example: artificial intelligence AND liver metastasis, machine learning AND liver metastases, radiomics AND liver metastases. AND shouldn’t be mentioned in Capital.

7.  There should be a separate section in the last part of the manuscript after Conclusion and before References about Contribution of the authors. Lines 86-87 should be moved to that location with addition of contribution of other authors as well.

8. Language correction proposed for Line 89-93. (Very Long and Unstructured Sentence).

Data extracted are algorithms used for CRLM prediction, number of patients and the 89 primary tumor they were affected by, diagnostic images used for prediction (CT or MRI), 90 division of the dataset for machine training (e.g. training set, test set and validation set) 91 and main parameters to recognize algorithm utility such as accuracy, specificity, recall 92 and precision.

9. Why study by Biglarian et al. 2012 was included in Table-1 as the review was performed from 2019-2022 as discussed in line 24. Update figure 1 accordingly as the number of studies should be 16 now and the number of patients should also be less than 14,475.

10. The objective of conducting this review is not clear. The Rationale behind the study should be emphasized. The writing is not clear.

11. Include CT/MRI images under different cross-sections and compare CT vs MRI. Also, discuss the evolution/viability of one field over another.

12. In MRI or CT, the three-dimensional information of blood vessels and delineation of irregular tumor boundaries is feasible while the tumor is lying deeper inside the human body as often experienced in clinical treatments, tumors may not be available for resection to perform histological analyses [http://hdl.handle.net/11603/25299]. Also, tumor margins are critically important as blood perfusion features representative of micro capillaries may be extracted in MRI also [https://doi.org/10.1016/j.cmpb.2020.105781]. The suggested algorithm has not been tested on MRI or CT or µCT data. Also, in case, capillaries or blood vessel level information is missing, blood perfusion extraction from the voxels may serve the purpose of information extracted after segmentation. Recent studies suggest that blood perfusion in tumors is anisotropic (heterogeneous in nature). Discuss this under Discussion section.

13. Rewrite Conclusion section with clear emphasis on key points.

Please proof-read the submission to correct the proposed changes. Spacing issues were noted with several long and unstructured sentences.

Author Response

The study “Artificial Intelligence to early predict liver metastases: a narrative review of 14,475 patients” by authors Avella et al. lies within the Journal scope of Life. There are several suggestions for authors to incorporate before recommending or reconsider the work for publication. The authors fail to formulate a Research Question? The authors’ focus was little shifted in writing. The authors should pay attention to all aspects and given comments in next submission and ensure that all points should be adequately addressed.  

  1. The title of the study should be revised. There should not be any mention of quantitative number in the title. Furthermore, “a narrative” should be “a narrative” for present title.

Thank you for your kind suggestion. We modified the title: the current form is “Artificial Intelligence to early predict liver metastases in patients affected by Colorectal Cancer: a narrative review of current status and future prospective.”

  1. The Abstract section should be rewritten. There are several long unstructured sentences that needs revision. For example: Due to higher information contents and 19 different procedures, it was not founded the greatest methodto predict LM so far. Revise Abstract section.

Thank you again. We revised the Abstract section according to your suggestion.

  1. Proof-read of entire manuscript is required.

For example in Introduction section there is no spacing between words or extra spacing within two words: (Lines 37-39), Line 42, Line 52, Line 62, Line 96, Line 130, Line 173.

In the last decades, multidisciplinary teams of surgeons, anesthesiologists, oncologists, and radiologists have beencollaborated to increase the oncologic patients’ Overall Survival (OS) and Disease-Free Survival (DFS).

Thank you, we revised the text.

What do you mean by trough matrices?

We modified the text with “through matrices” according you observations.

Line 62: theadvantages

We modified the text.

Line 173: Plural: For these reason

We modified the text.

  1. The Introduction section should be rewritten. In current submission, the provided Introduction is very weak.

Thank you again for your suggestions. We modified the section in order to improve the quality of Introduction paragraph.

  1. Section 2 name should only be “Materials and Methods” and not Materials and Methods Literature Research.

Thank you for your suggestion: we reviewed the Materials and Methods section.

  1. Criterion used to hunt for studies should be better described. Also, the authors are advised to pay attention on punctuation. For example: artificial intelligence AND liver metastasis, machine learning AND liver metastases, radiomics AND liver metastases. AND shouldn’t be mentioned in Capital.

Thank you again. We reported the keywords used to perform articles research in PubMed DataBase.

  1. There should be a separate section in the last part of the manuscript after Conclusion and before References about Contribution of the authors.

Lines 86-87 should be moved to that location with addition of contribution of other authors as well.

We reported the author contributions next to conclusion section. Thank you for your suggestion.

  1. Language correction proposed for Line 89-93. (Very Long and Unstructured Sentence).

Data extracted are algorithms used for CRLM prediction, number of patients and the 89 primary tumor they were affected by, diagnostic images used for prediction (CT or MRI), 90 division of the dataset for machine training (e.g. training set, test set and validation set) 91 and main parameters to recognize algorithm utility such as accuracy, specificity, recall 92 and precision.

According to your suggestion, we modified the sentences in order to clarify data extraction phase.

  1. Why study by Biglarian et al. 2012 was included in Table-1 as the review was performed from 2019-2022 as discussed in line 24. Update figure 1 accordingly as the number of studies should be 16 now and the number of patients should also be less than 14,475.

Dear Reviewer, we corrected the timeline included in our analysis (from 2019 to 2022 to 2012 to 2022) according to our review. Furthermore, we recalculate the number of patients included in our analysis. Thank you.

  1. The objective of conducting this review is not clear. The Rationale behind the study should be emphasized. The writing is not clear.

Thank you again. We modified the text.

  1. Include CT/MRI images under different cross-sections and compare CT vs MRI. Also, discuss the evolution/viability of one field over another.

Thank you again. We added CT images of liver parenchyma analysis and Regions of Interest.

  1. In MRI or CT, the three-dimensional information of blood vessels and delineation of irregular tumor boundaries is feasible while the tumor is lying deeper inside the human body as often experienced in clinical treatments, tumors may not be available for resection to perform histological analyses [http://hdl.handle.net/11603/25299]. Also, tumor margins are critically important as blood perfusion features representative of micro capillaries may be extracted in MRI also [https://doi.org/10.1016/j.cmpb.2020.105781]. The suggested algorithm has not been tested on MRI or CT or µCT data. Also, in case, capillaries or blood vessel level information is missing, blood perfusion extraction from the voxels may serve the purpose of information extracted after segmentation. Recent studies suggest that blood perfusion in tumors is anisotropic (heterogeneous in nature). Discuss this under Discussion section.

Thank you again for your comments. We analyzed this topic in discussion section.

  1. Rewrite Conclusion section with clear emphasis on key points.

Thank you. We reviewed our paper according to your suggestions.

Reviewer 3 Report

The authors performed a thorough review of utilizing AI on radiographic images for early diagnosis of liver mets in mostly CRC.  The authors selected 17 studies over a 3-4 year period between 2019 and 2022.  Several different AI analyses were used with an accuracy of about 90%.  Using radiomics and AI, the authors demonstrated that it is an effective method in a clinical setting.

The english language is fine.  There are minor grammatical issues that need to be addressed.  For example, there are several instances where words are combined.   A thorough reading should fix these issues. 

Author Response

The authors performed a thorough review of utilizing AI on radiographic images for early diagnosis of liver mets in mostly CRC.  The authors selected 17 studies over a 3–4-year period between 2019 and 2022. 

Several different AI analyses were used with an accuracy of about 90%. 

Using radiomics and AI, the authors demonstrated that it is an effective method in a clinical setting.

The english language is fine. 

There are minor grammatical issues that need to be addressed. 

For example, there are several instances where words are combined. 

A thorough reading should fix these issues.

Thank you for your kind consideration and opinions. We modified the text according to your suggestions. 

Round 2

Reviewer 2 Report

Peer Review Report

Manuscript ID: Life-2626085

Title: Artificial Intelligence to early predict liver metastases in patients affected by Colorectal Cancer: a narrative review of current status and future prospectives

The study “Artificial Intelligence to early predict liver metastases in patients affected by Colorectal Cancer: a narrative review of current status and future prospectives” is considerably improved following first peer review. However, the manuscript still needs improvement. The coherence is missing in writing. The authors should pay attention to all aspects and given comments in next submission and ensure that all points should be adequately addressed.  

1. The title of the study should be revised. The title is very long.

2. There is inconsistency in font size and font type across the manuscript. Certain portion of the manuscript was written in Book Antique while other in Times New Roman. Correct such changes.

3. Line 32-33, Conclusion should start from new line. Line 29, there should be space between “3cases” as “3 cases”.

4. The study missing discussion on blood perfusion heterogeneity of tumors extracted from MRI images [https://doi.org/10.1016/j.ijheatmasstransfer.2023.124698] which should be addressed in present context of work.

5. Furthermore, we advise the authors to avoid contradicting language: Lines 101-102

No articles were available in other databases. The items discarded are summarized in Figure 1.

6. Why figure 2 is in Discussion section? The accurate location will be somewhere in context of methodology or as a brief introduction. Figure 3 should be in Results section. Any result should be in Results section.

7. Line 344: Healthy and no-healthy patients. “No-healthy patients” is not a correct usage. It should be diseased or similar.

8. Revise Limitations section.

9. Why there is space between Line 351-353 and 360-363. There is inconsistent space throughout the manuscript. Please remove the excessive space.

10. Emphasize what novelty you bring to the field after doing this review? What was unknown? What is the research question? How do you address the problem?

Coherence is missing in the writing.

Author Response

Dear Editor,
thank you for considering our paper for publication in your eminent journal.

Please find hereby enclosed our corrections in order to answer to Reviewer’s comments on the manuscript entitled

Artificial Intelligence to early predict liver metastases: a narrative review of 14,475 patients.

By Pasquale Avella1,2*, Micaela Cappuccio2*, Teresa Cappuccio3, Marco Rotondo3, Daniela Fumarulo3, Germano Guerra3, Guido Sciaudone3, Antonella Santone3, Gianfranco Camilleri4, Paolo Bianco1 and Maria Chiara Brunese3

that we submit for publication in Life, MDPI.

We have reviewed our paper according to Editor and Reviewers suggestions and we summarized below point by point the analysis made:

Reviewer 2

The study “Artificial Intelligence to early predict liver metastases in patients affected by Colorectal Cancer: a narrative review of current status and future prospectives” is considerably improved following first peer review. However, the manuscript still needs improvement. The coherence is missing in writing. The authors should pay attention to all aspects and given comments in next submission and ensure that all points should be adequately addressed.  

  1. The title of the study should be revised. The title is very long.

Thank you for your suggestions. We reviewed our manuscript title according to Reviewer 1 comment. Furthermore, we modified the title to reduce the number of words. We hope that this version is welcomed by both reviewers.

  1. There is inconsistency in font size and font type across the manuscript. Certain portion of the manuscript was written in Book Antique while other in Times New Roman. Correct such changes.

We have changed the font type font size according to the suggestion.

  1. Line 32-33, Conclusion should start from new line. Line 29, there should be space between “3cases” as “3 cases”.

Thank you. We modified the text.

  1. The study missing discussion on blood perfusion heterogeneity of tumors extracted from MRI images [https://doi.org/10.1016/j.ijheatmasstransfer.2023.124698] which should be addressed in present context of work.

We analyzed the current issued in the Limitations section.

  1. Furthermore, we advise the authors to avoid contradicting language: Lines 101-102

No articles were available in other databases. The items discarded are summarized in Figure 1.

Dear reviewer, thank you again. We changed the sentence to “No studies were available in the Cochrane Library, Kaggle and Data.gov databases.  Figure 1 summarizes the number of papers and related exclusion criteria after title, abstract and full-text analysis.”

  1. Why figure 2 is in Discussion section? The accurate location will be somewhere in context of methodology or as a brief introduction. Figure 3 should be in Results section. Any result should be in Results section.

We included Figure 2 in the discussion section to better analyze the different AI methods reported in the literature. Moreover, the presentation order has been modified by us according to your suggestions.

  1. Line 344: Healthy and no-healthy patients. “No healthy patients” is not a correct usage. It should be diseased or similar.

Thank you again. We modified the text to “unhealthy patients”.

  1. Revise Limitations section.

We revised and deployed the limitations section. Thank you.

  1. Why there is space between Line 351-353 and 360-363. There is inconsistent space throughout the manuscript. Please remove the excessive space.

Thank you. We removed the excessive space.

  1. Emphasize what novelty you bring to the field after doing this review? What was unknown? What is the research question? How do you address the problem?

We reported any considerations of our review data in the “current status and future prospectives” and “limitations” sections.

Pasquale Avella,

Department of Medicine and Health Sciences "V. Tiberio",
University of Molise, Campobasso, Italy,
Via Francesco De Sanctis, 1, 86100 Campobasso CB,
e-mail: avella.p@libero.it